# Acoustic analysis of bottlenose dolphin vocalizations for behavioral classification in controlled settings

Laura Screpanti[1], Francesco Di Nardo[1]*, Rocco De Marco[2], Stefano Furlati[3], Giacomo Bucci[3], Alessandro Lucchetti[2,4], David Scaradozzi[1,4,5]

1 Department of Information Engineering, Università Politecnica delle Marche, Ancona, Italy, 2 Institute of Biological Resources and Marine Biotechnology (IRBIM), National Research Council (CNR), Ancona, Italy, 3 Oltremare Marine Park, Riccione, Italy, 4 National Biodiversity Future Center, Palermo, Italy, 5 ANcybernetics, Università Politecnica delle Marche, Ancona, Italy

* f.dinardo@staff.univpm.it

## Abstract

Understanding how bottlenose dolphins adjust their vocal behavior in response to daily routines can provide insights into social communication and welfare assessment in managed care environments. This study presents a detailed analysis of bottlenose dolphin (*Tursiops truncatus*) vocal behavior in relation to different daily activities within a controlled environment at Oltremare Marine Park (Riccione, Italy). 24 hours of continuous acoustic recordings were collected from seven dolphins during a typical day at the marine park, including training, feeding, playing, and unstructured activities. Signals were analyzed to quantify the variations in type and number of vocalizations in relation to dolphin activity. 3,111 whistles were manually extracted and stored as both normalized audio files and high-resolution spectrograms. Additionally, an automated algorithm identified 1,277 pulsed vocalizations, classified into echolocation click trains, burst-pulse sounds, and feeding buzzes, using signal-to-noise ratio (SNR) and inter-click interval criteria. Results revealed a significant increase in vocalization rates during structured activities compared to unstructured periods ($p < 0.001$). This trend was consistently observed across all four vocalization types. Notably, play sessions elicited the highest rates of pulsed vocalizations ($p < 0.01$), suggesting enhanced social and exploratory behaviors, i.e., interactions with conspecifics as well as curiosity-driven engagement with the environment. To test dataset reliability and usability, signal quality was analyzed by evaluating SNR. To support future research in marine mammal bioacoustics, behavioral ecology, and Artificial-Intelligence-based acoustic monitoring, the full annotated dataset was released as an open-access resource.

**Data availability statement:** The data that support the findings of this study are deposited in the public repository SEANOE and are freely downloadable at the following link: https://doi.org/10.17882/109081 The full citation of the dataset is: Di Nardo F, De Marco R, Scaradozzi D. Bottlenose Dolphin Vocalizations in Controlled Environments: A Dataset for Behavioral Classification. SEANOE. 2025. https://doi.org/10.17882/109081.

**Funding:** This work was supported in part by LIFE Financial Instrument of the European Community, Life Delfi Project – Dolphin Experience: Lowering Fishing Interactions (LIFE18NAT/IT/000942) and by the National Recovery and Resilience Plan (NRRP), Mission 4 Component 2 Investment 1.4 (Call for tender No. 3138 of 16 December 2021, rectified by Decree n.3175 of 18 December 2021 of Italian Ministry of University and Research funded by the European Union) NextGenerationEU. The study was made possible through the support and collaboration of Costa Edutainment, which provided access to their Riccione facility. Special acknowledgment is given to Barbara Marchiori, Gianni Bucci, Barbara Acciai, Paola Righetti, and Claudia Di Mecola for their dedicated contributions and support during the project. The funders had no role in study design, data collection and analysis, decision to publish, or preparation of the manuscript. There was no additional external funding received for this study.

**Competing interests:** The authors have declared that no competing interests exist.

## 1. Introduction

Understanding the vocal behavior of dolphins is essential for assessing their communication strategies, social interactions, and responses to environmental and anthropogenic factors [1]. Over the past few years, monitoring marine ecosystems has become increasingly central to research and conservation efforts. Among the available methodologies, Passive Acoustic Monitoring (PAM) has emerged as a particularly effective and non-invasive strategy [2,3]. This technique employs hydrophones, i.e., underwater microphones, along with appropriate recording equipment to capture and analyze dolphin vocalizations [4]. Such an approach enables scientists to detect the presence and behavior of these animals without the need for physical proximity or disturbance. PAM proves especially useful in offshore, deep-water, or otherwise inaccessible locations and under adverse weather conditions where visual surveys are not feasible. Through the provision of continuous, high-quality acoustic data, PAM contributes significantly to conservation practices by supporting the identification and reduction of anthropogenic pressures, including acoustic pollution, habitat disruption, and harmful interactions with fisheries [5]. One of the main purposes of underwater monitoring is the collection, storage, analysis, and interpretation of marine bioacoustic signals. Recent advancements in acoustic data recording and collection have enabled researchers to capture extensive datasets of dolphin sounds and provide them as publicly available resources, offering unprecedented opportunities to analyze their vocal repertoires in detail [6–11].

Dolphins produce a range of vocalizations, including whistles, echolocation clicks, burst-pulse sounds, and feeding buzzes, which are used for echolocation, social bonding, and coordination within groups [12]. In this context, social bonding refers to affiliative behaviors that strengthen relationships within the group and coordination with groups indicates the alignment of actions among dolphins during collective activities such as hunting or traveling. The above-mentioned vocalizations are highly adaptable, varying with social context, environmental conditions, and external stimuli, reflecting the dolphin's cognitive complexity and behavioral flexibility. Whistles are generally regarded as communicative signals, with their production increasing during social interactions among dolphins. These vocalizations are commonly associated with functions such as individual recognition, coordination of collective behaviors, and regulation of group movements. Typically, they span frequencies from approximately 1–25 kHz and exhibit durations ranging from 0.1 seconds to few seconds [11]. In contrast, echolocation clicks are short, broadband acoustic pulses that can reach frequencies up to 140 kHz. These signals play a fundamental role in navigation and prey detection, allowing dolphins to generate detailed auditory representations of their environment [13]. Burst-pulse vocalizations consist of fast sequences of acoustic pulses or clicks with similar characteristics to echolocation clicks. They are distinguished by their high repetition rate and variable spectral characteristics. These vocalizations are believed to serve social functions and are frequently observed in contexts involving agonistic behavior, such as during competitive feeding events [14]. Feeding buzzes are rapid sequences of pulses emitted by dolphins as they close

in on prey. These high-rate pulse trains occur during the final phase of prey pursuit, when the animal requires fine-scale spatial information to catch its moving target. For bottlenose dolphins, dominant click frequencies typically fall between 40 and 80 kHz [15].

Research into the influence of activity type on dolphin vocalizations has shown that structured activities, such as training sessions and play interactions, can elicit distinct acoustic patterns compared to unstructured free periods [16]. Understanding this phenomenon is critical for gaining insight into how dolphins modulate their vocal behavior in response to different conditions, which could have direct implications for their welfare and the management of their environments. However, the extent to which vocalization patterns differ between these contexts remains insufficiently explored. Moreover, there is very limited availability of high-quality datasets in marine bioacoustics, which further complicates achieving reliable analyses. The availability of these datasets would be essential for advancing research on dolphin communication and behavior. Open-access datasets would enable researchers worldwide to analyze and compare vocalization patterns across different environments, fostering a deeper understanding of cetacean acoustics. By sharing comprehensive datasets, the scientific community could enhance the development of new analytical methods, improve the accuracy of automated detection techniques, and contribute to the conservation and welfare of marine mammals.

Thus, this study has a twofold aim: first, to present a detailed analysis of the vocal behavior of seven bottlenose dolphin (*Tursiops truncatus*) in relation to different daily activities within a controlled environment at Oltremare Marine Park (Riccione, Italy); second, to share the labelled acoustic dataset collected for the present analysis across various activity contexts. The analysis aimed to contribute to a deeper understanding of dolphin communication and provide practical insights for designing enrichment activities and optimizing welfare practices for bottlenose dolphins in managed care. The release of dataset is expected to serve as a significant resource for the marine bioacoustics community, offering a foundation for further investigations into dolphin behavior, ecology, and interaction with human activities. In the Mediterranean Sea, several dolphin species can be found, each with distinct features and behaviors. The Striped Dolphin (*Stenella coeruleoalba*) is sleek and agile, with gray and white stripes radiating from the eyes, often seen in large pods performing energetic leaps and acrobatics. The Common Dolphin (*Delphinus delphis*), once widespread but now rare, is recognizable by its striking yellowish hourglass pattern along the sides and its dynamic group behavior. The Bottlenose Dolphin (*Tursiops truncatus*) is robust and gray with a curved dorsal fin and a short, broad snout, known for its intelligence and frequent interactions with humans. The Risso's Dolphin (*Grampus griseus*) stands out with its rounded head and heavily scarred body, which becomes paler with age, and typically lives in smaller, more reserved groups. The bottlenose dolphin is the most commonly represented cetacean species in European zoological facilities. Therefore, this dolphin species was chosen for this study, based on the opportunity to simply record vocalizations in a nearby facility that offered suitable experimental conditions.

Whistles are the most extensively studied category among the bottlenose dolphin vocalizations, particularly in social behaviors [10,17–22]. Whistle characteristics are known to vary according to context, such as activity state, group size and composition, geographic location, and ambient noise levels [16]. In captive environments, whistles are closely tied to dolphin activities and interactions [23]. For example, whistles emitted during feeding are typically shorter in duration and higher in frequency compared to those produced during other activities [17]. Conversely, longer whistle duration or irregular patterns can indicate stress, discomfort, or agitation [21,22]. Whistle production in training sessions may reflect engagement and responses to reinforcement cues [19], while social interactions can elicit affiliative whistles conveying playfulness or greetings [18]. The role of experience and familiarity has also been explored, showing that visual attention and behavior in the presence of humans are shaped by these factors [24,25]. Recent studies have expanded the understanding of context-dependent vocalizations, revealing unexpected repertoire variations influenced by environmental and social factors [26].

Given the well-documented significance of whistles in bottlenose dolphin communication, the current dataset has been specifically designed to provide a comprehensive collection of all whistle vocalizations identified by an expert PAM analyst

during the experimental campaign. These whistle recordings have been meticulously extracted and stored in the form of spectrograms, ensuring a high level of detail and accessibility for further examination. To complete the analysis of the dataset, the main pulsed vocalizations (echolocation clicks, burst-pulse sounds, and feeding buzzes) were also identified using an automated algorithm. A statistical analysis was then conducted on the identified whistles and pulsed vocalizations to better highlight the characteristics of the dataset. This approach enables a deeper investigation into how these acoustic signals correlate with different dolphin activities throughout the day.

## 2. Materials and methods

### 2.1. Data collection

Data was recorded at the Oltremare Marine Park in Riccione (Italy), which hosted 7 bottlenose dolphins (2 males and 5 females) in the "Laguna dei Delfini" facility. Dolphins can swim in different interconnected environments. Each environment can be separated from the other employing special partitions, so that the specimens, or groups of them, can be confined in one or more compartments in case of management or veterinary needs. The main environments are schematized in Fig 1. The main activities carried out by the trainers were noted, also in connection with the distribution of food happening during this kind of activity. The food, which consists of medium-sized fish (sardines, suri, etc.), was either thrown into the pool or put directly into the mouth.

The sampling session ran continuously, starting on 11/20/2021 at 10:15 a.m. and ending on 11/21/2021 at 10:30. The main objective of the exercises conducted by trainers and the tasks assigned to the dolphins is to promote both physical and cognitive stimulation, as well as to enhance the variability of the controlled environment, which is inherently less stimulating than the natural habitat. These activities are not specifically aimed at enhancing vocal behavior. In particular, vocal conditioning and the reinforcement of vocalizations are not considered integral components of the training protocol. All trainers possessed substantial professional experience, ranging from a minimum of 5 up to 10 years. The operational protocol mandates that all trainers engage uniformly with each individual dolphin, a practice specifically designed to prevent the emergence of preferential bonds and to maintain balanced human-animal interactions across the group. The dolphin group housed at the facility had remained stable for several years, ensuring consistency in social dynamics and behavioral baselines.

Data were collected with the approval and under the supervision of Oltremare staff. No additional permits were required, as all data were gathered non-invasively using PAM equipment that emitted no sound or light and involved no direct interaction with the dolphins. The study did not alter the dolphins' routine, which remained under the facility staff's standard care. Equipment was installed and concealed in a hatch at the bottom of the pool, while dolphins were absent.

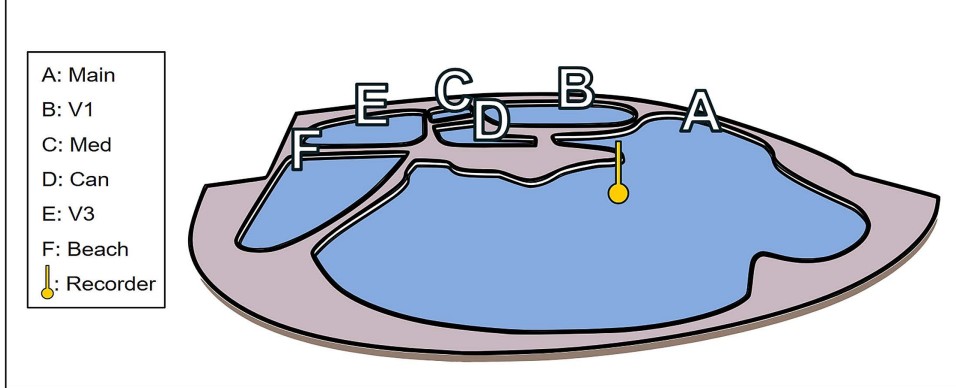

**Fig 1. The "Laguna dei delfini" pool at the Oltremare park in Riccione (Italy).**

Equipment was sterilized prior to use, in compliance with Marine Park regulations. Animal care followed institutional guidelines.

## 2.2. Dolphins activities

During the sampling sessions, trainers adhered to the schedule outlined in Table 1, which details the timing, duration, and type of activities conducted during the monitored period. The distribution of activities included both structured training and unstructured interactions, with variability in frequency and duration. The use of positive reinforcement and the inclusion of rest periods between sessions were designed to support sustained engagement and promote animal welfare throughout the study. Ordinary sessions (ORD) involved standard training exercises where dolphins received positive reinforcement, such as food rewards or other preferred stimuli, upon completing tasks. Typically, these sessions lasted between 30 and 40 minutes.

The play session (PLAY) encouraged spontaneous and interactive behaviors among the dolphins. During these sessions, trainers introduced floating toys, such as inflatables, into the pool, allowing the dolphins to engage freely. This session occurred once during the sampling period and lasted approximately 40 minutes. Unlike ORD sessions, food rewards were not provided. A specialized session, named "fish from the roof" (FFR), was conducted for experimental purposes. In this session, additional fish were released into the pool near the hydrophone to promote specific behaviors relevant to the study. The FFR activity lasted approximately 40 minutes. In addition to the guided activity phases, two other phases are considered: the nighttime phase (NIGHT) and the daytime phase of free dolphin activity (FREE ACT). The NIGHT phase is characterized by the absence of any interaction between dolphins and trainers, as all staff members have left the facility. This phase lasts from 7:00 P.M. on November 20–7:00 A.M. on November 21. Given the lack of trainer interaction and the nighttime setting, it is assumed that the number of dolphin vocalizations is minimal. Therefore, this phase is used as a baseline reference for comparison with the guided activity periods. The FREE ACT phase, on the other hand, takes place during daylight hours when dolphins are freely active within the facility without any structured or guided interaction with trainers. However, due to the highly unpredictable nature of dolphin behavior during this period, vocalizations recorded during the FREE ACT phase cannot be reliably associated with specific activities.

## 2.3. Recording technology

Acoustic measurements were carried out thank to the Dodotronic and Nauta's UREC U384K autonomous underwater recorder (sampling rate = 192 kHz; resolution = 16 bits; bandwidth = 10 Hz–96 kHz; gain = 10x (20 dB)) [27] equipped with a preamplified hydrophone Sensor Technology SQ26–08 [28], which declares a sensitivity of −193.5 dB re 1 V/μPa @ 20ºC at 1 m between at least 1 Hz and 28 kHz. However, further studies tested its sensitivity at higher frequencies, reporting a sensitivity of −169 dB re 1 V/μPa up to 50 kHz for the same combination of hydrophone and recorder [29]. The low-frequency response was not restricted during recording.

**Table 1. Dolphins' activities throughout the recording phase: Ordinary activity (ORD); Play activity (PLAY); Fish from the roof (FFR).**

| Date | Start | End | Activity |
|------|-------|------|----------|
| 20/11 | 10:20 | 11:00 | ORD |
| | 12:00 | 12:45 | ORD |
| | 14:45 | 15:15 | ORD |
| | 15:20 | 16:00 | PLAY |
| | 16:05 | 16:45 | FFR |
| | 16:50 | 17:25 | ORD |
| 21/11 | 9:30 | 10:00 | ORD |

The UREC U384K is presented as a small device approximately 35 cm in length and 13 cm in diameter, including a protective cover in place to repair the hydrophone. It is powered by 3 D-type alkaline batteries and has an autonomy of more than 72 hours of continuous recording activity. The UREC U384K has an internal RTC that can be synchronized via a smartphone app, thus ensuring effective temporal correlation of events within the recordings. The device stores recordings in 5-minute.wav files, each marked with a block start date and time. The recording device was placed at the bottom of the main pool (Fig 2), which has an area of 1,173 m², a capacity of 5,606 m³, and a maximum depth of 6 meters.

The device was placed inside a hatch (Fig 1) in order to avoid direct interaction with the animals and placed in a barycentric position both in relation to the pool and in relation to the activities carried out by the trainers.

## 2.4. Whistle identification

To perform the whistle identification, each 5-minute file was visualized within a 4-second timeframe using Audacity's spectrogram interface (accessible at www.audacity.org), with a Hann window comprising 1,024 points. Fig 3 provides an example of the visualization available to the PAM operator.

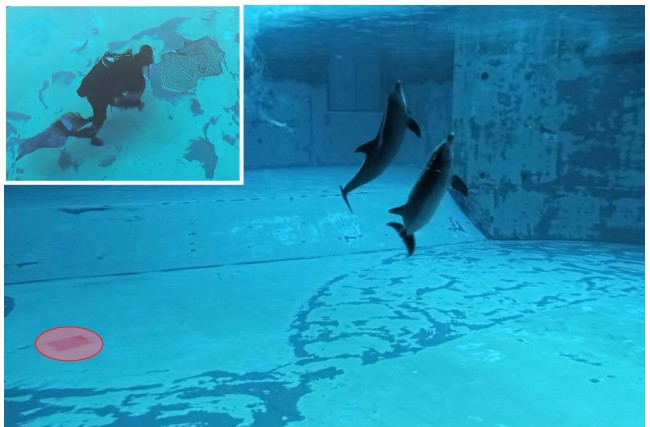

**Fig 2. Positioning of the recording device (red circle) in the "Laguna dei delfini" pool.**

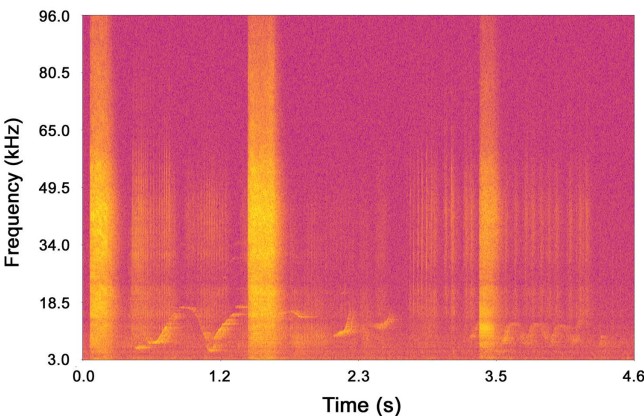

**Fig 3. Example of a spectrogram including multiple whistles and pulsed vocalizations.**

A trained and experienced passive PAM operator examined each audio spectrogram to detect whistles. Whistles were identified through visual inspection and manually centered within the 4-second window. Each whistle was then isolated, trimmed to its effective duration, saved as a WAV file, and stored in the dataset. On the 4-second spectrogram, a whistle could appear as a continuous shape, even if it comprised multiple consecutive short segments. Instances of multiple whistles occurring simultaneously from different dolphins within the same timeframe were noted, along with the start and end times of each whistle, and their durations were computed. Additionally, the number of whistles in each window was tallied, and their positions (start and end) were reported. Then, whistles were classified based on their duration ($d$) into the following classes:

- Class 1: $d \leq 0.2$ s
- Class 2: $0.2 < d < 0.4$ s
- Class 3: $0.4 < d < 0.8$ s
- Class 4: $d \geq 0.8$ s

The classification of whistles based on their duration was instrumental in gaining insights into the behavioral context of vocalizations. Whistle duration is often indicative of the underlying function or social context; for example, short-duration whistles may reflect alertness or rapid interactions, while long-duration whistles are typically associated with social bonding, group cohesion, or complex communication. This categorization aims to support the examination of potential relationships between whistle duration and specific activities during the recording sessions, facilitating a more detailed analysis of vocalization patterns

## 2.5. Analysis of high-frequency pulsed vocalizations

To perform the identification of pulsed vocalizations, a basic automated algorithm was employed following the guidelines reported in [30]. This approach focuses on the signal-to-noise ratio (*SNR*) to identify the pulsed vocalizations and on the inter-click interval (*ICI*) to distinguish between the three different vocalizations. *ICI* is the absolute time distance between two consecutive peaks. The first step consists of computing the mean *SNR* value along 2-ms windows throughout the whole signal. Then, the occurrence of a peak was identified in a specific 2-ms window when the following two requirements are met simultaneously:

- mean *SNR* $\geq$ *Th*;
- mean SNR values in the previous and successive 2-ms windows $<$ *Th*.

*Th* is a specific threshold value determined through a sensitivity analysis of the number of detected peaks to the *SNR* threshold value. Then, the detected peaks were classified into three categories, based on the fact that peak sequences exhibit different *ICI* values [31]:

- *Echolocation Click Trains (ECT)*: peak sequences with an average *ICI* longer than 0.22 s.
- *Burst-Pulse Sounds (BPS):* peak sequences with an average *ICI* between 0.017 and 0.22 seconds.
- *Feeding Buzzes (FB):* peak sequences with an average *ICI* shorter than or equal to 0.017 seconds.

The SNR was computed for each signal sample as follows: firstly, the signal was high-pass filtered (Butterworth filter, fifth order, cut-off frequency: 3 kHz) using the Audacity free audio editor (www.audacity.org) to remove the background noise from different sources; then the following formula was computed:

$$SNR(t) = 10 \ \log_{10} \cdot \frac{\sigma^2_{signal}}{\sigma^2_{noise}}$$

(1)

where *noise* is computed as the signal value in a sample window where no dolphin vocalizations were detected. This segment was identified through visual inspection. This automated method allowed for a synthesized overview of the pulsed vocalizations, facilitating a broad characterization of the acoustic data. The basic algorithm code developed here for peak detection is available at the figshare general-purpose open repository [32]. While the *ICI*-based approach effectively differentiates among ECT, BPS, and FB categories, it is acknowledged that more detailed and precise methods could be applied for further studies or in-depth analyses. From the pre-processed recordings, the total counts of whistles, ECT, BPS, and FB were extracted and saved into a MS Excel spreadsheet (see section II.G).

## 2.6. Data analysis in specific behavioral activities

A further analysis was conducted to quantify the occurrence of dolphin vocalizations associated with the various activities performed by the animals throughout the day at the marine park (ORD, PLAY, and FFR) and during free activities without external stimuli in the park closed hours (7:00 p.m. – 7:00 a.m. of the following day, NIGHT). Specifically, based on the computations described in the previous sections, the occurrence of whistles associated with different activities was quantified, followed by a more detailed breakdown into the four whistle categories characterized by different durations, as described in Section II.D. The same approach was applied to pulsed vocalizations, both as a whole and by differentiating among ECT, BPS, and FB. Each of these quantitative assessments was supported by a thorough statistical analysis. To avoid confounding factors, the period from 5:00 a.m. to 7:00 a.m. was excluded from the analysis, due to disturbances from uncontrolled external events, which resulted in an unusually high number of clicks and whistles.

## 2.7. Data description

The raw recordings, the whistle spectrograms, the WAV files corresponding to each spectrogram, the vocalization characteristics, and the labels for identifying the vocalizations in the raw file have been made available via a repository at SEA-NOE [33]. The details of each file are reported below.

*Raw recordings:* Over 30 GB of continuous raw signals are released, amounting to 1,430 minutes. The raw signals are released in 19 separate folders, each containing 15 five-minute audio segments, to simplify downloading for users. Each segment has a duration of five minutes and is stored in standard uncompressed WAV format. The file name is in the format: YYYYMMDD_hhmmss_192.wav. The recording interval spans from 11/20/2021 at 10:21 to 11/21/2021 at 10:11.

*Whistle spectrograms:* The spectrograms of each whistle have been grouped into subfolders that share the same name as the 5-minute block from which they were extracted. The spectrogram was generated within the 0–25 kHz range, using a linear scale, NFFT = 1024, using a custom Python script which employs scipy (1.13.1), numpy (1.26.4), matplotlib (3.5.1), and pillow (10.3.0) libraries. The file name is the same as the original recording file in its first part (YYYYMMDD_hhmmss_192), followed by "-colspectro-W-OFFSET.png", where W stands for whistle and OFFSET indicates the time offset in seconds from the beginning of the individual recording from which it was extracted.

*Whistle WAV files:* The WAV files corresponding to each of the spectrograms have also been made available in sub-folders named after the original file from which they were extracted. The extracted signal was normalized within the range 0–1. The file name is the same as the original recording file in its first part (YYYYMMDD_hhmmss_192), followed by "-colspectro-W-OFFSET.wav".

*Vocalization characteristics:* A summary of vocalization characteristics has been compiled in an Excel spreadsheet named dataset_filtered.xlsx, made available alongside the dataset repository. This file includes structured information extracted from the raw dataset. Each row of the spreadsheet corresponds to a 5-minute audio segment, identified by its start timestamp in the format YYYYMMDD_hhmmss. For each segment, the file reports the total number of whistles and pulsed vocalizations (classified as ECT, BPS, and FB), along with the number of whistles grouped by duration class (Class 1 to Class 4, as defined in Section II.D). The spreadsheet also annotates the type of dolphin activity associated with each time block (e.g., ORD, PLAY, FFR, NIGHT), facilitating a rapid mapping between vocal events and behavioral contexts.

These annotations serve as a valuable resource for researchers aiming to perform additional correlation analyses or develop automated behavior classification tools. The tabular format of this summary ensures ease of access and interpretability, supporting both manual inspection and algorithmic processing.

*Labels:* The timing data for each detected vocalization are provided as tab-separated.txt files, including:

1. Vocalization start time expressed in seconds as the time stamp in the original full recording, counting from the start of the recording;

2. Vocalization end time expressed in seconds as the time stamp in the original full recording, counting from the start of the recording;

3. Classification tag.

All the parameters reported in these files are also depicted as labels that can be imported directly into the Audacity software for visualization (file ◊ import ◊ labels).

## 2.8. Statistics

To assess the statistical significance of differences between two conditions, a Mann-Whitney U test was performed, given its suitability for comparing non-parametric data distributions. This analysis provided an initial evaluation of whether vocalization patterns differed meaningfully between the structured and unstructured behavioral states. Further analysis was conducted to investigate statistically significant differences in vocalization patterns among the free activity period and the organized activities (ORD, PLAY, and FFR). Specifically, the analysis focused on total vocalization counts, ECT, FB, BPS, and the classified categories of whistles based on their duration. The Kruskal-Wallis test was employed to evaluate these differences. Post hoc comparisons were performed using Dunn's test with Bonferroni correction to identify specific group-level differences [34]. The data analysis was conducted using R version 4.4.1 within the RStudio 2024.04.2 environment.

## 3. Results

Table 2 reports the total number, the duration, and the SNR value for each dolphin vocalization considered in the present study. Values are reported as mean ± standard deviation (SD).

The distribution of occurrences of whistles and pulsed vocalizations over time is illustrated in Fig 4 and Fig 5, respectively. Each bar corresponds to the number of vocalizations that occurred in a 5-minute block. These figures also indicate the timing of the three different typologies of activity sessions (ORD, FFR, and PLAY) as well as the NIGHT period. In gray, the period is shown when no organized activity was scheduled, but the dolphins were free to move and interact as they wished (FREE ACT). The vocalizations emitted during FREE ACT were not included in the statistical analysis because the dolphin behavior in this free phase is completely unpredictable, and it is not reliable to be used as a reference for statistical analysis. Fig 4 and Fig 5 provide an overall view of the increase in the number of vocalizations

**Table 2. SNR values.**

| Vocalization Type | Number | Duration (s) | SNR (dB) |
|---|---|---|---|
| Whistle | 3111 | 0.63 ± 0.45 | 11.9 ± 7.3 |
| ECT | 517 | 1.63 ± 0.63 | 6.2 ± 3.3 |
| BPS | 489 | 1.27 ± 0.49 | 9.4 ± 5.9 |
| FB | 271 | 0.69 ± 0.32 | 13.6 ± 6.2 |
| Mean | | 0.82 | 11.1 |
| SD | | 0.59 | 7.0 |

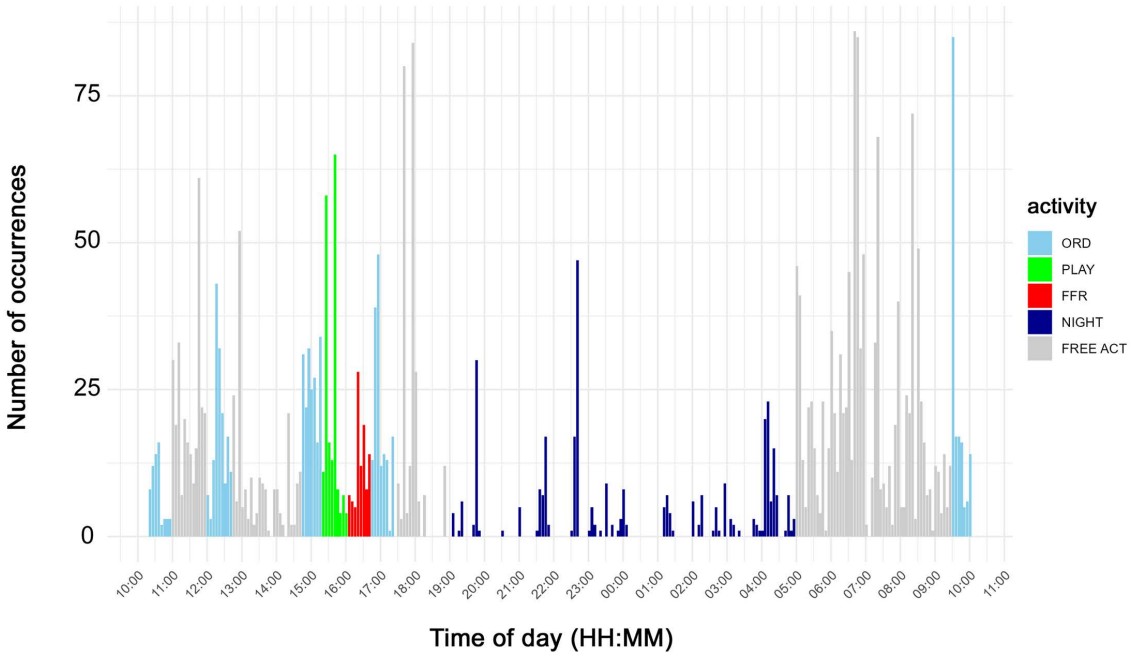

**Fig 4. Whistle occurrences over time.**

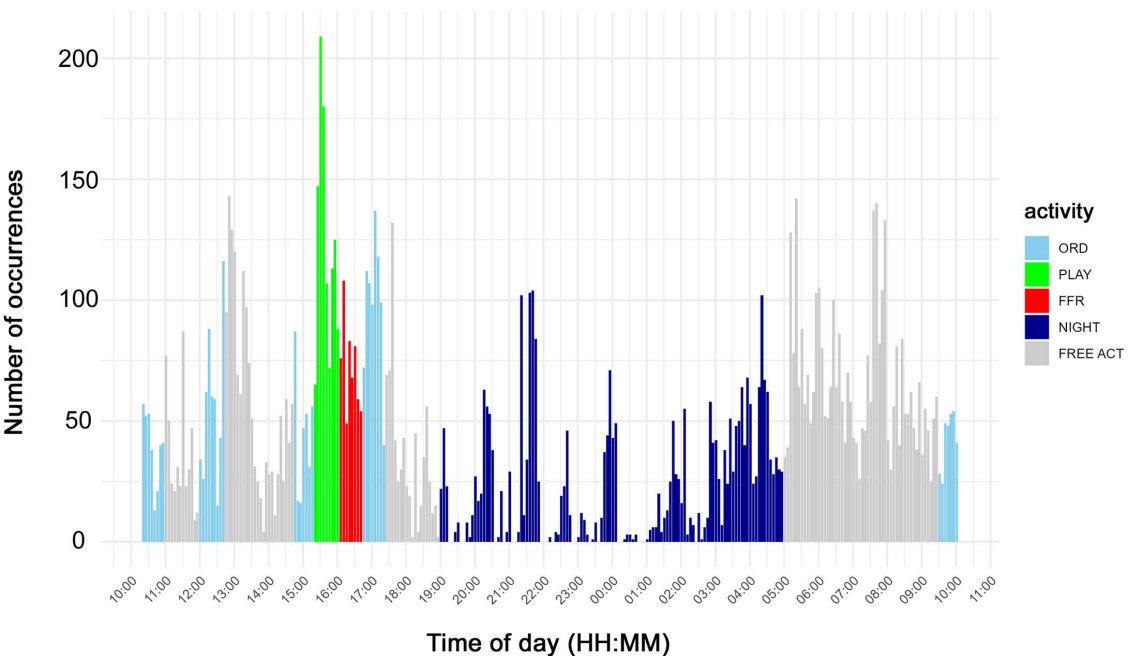

**Fig 5. Occurrences of pulsed vocalization over time.**

during periods of organized activity compared to the nighttime period. This observation applies to both whistles (Fig 4) and pulsed vocalizations (Fig 5). The fact that this phenomenon is observed despite the shorter duration of the organized activity period, when compared to the nighttime period, further strengthens the validity of the result. This outcome about whistles is confirmed by the statistical analysis. The Mann-Whitney U test, indeed, showed that the total number of whistles significantly increased during the global organized activity (ORD, FFR, and PLAY) compared to NIGHT period ($p<0.001$). This increase remains significant even when considering the various classes based on whistle duration: class 1 ($p<0.001$), class 2 ($p<0.001$), class 3 ($p<0.001$), and class 4 ($p<0.0001$). Similarly, the total counts of pulsed vocalizations were significantly higher during activity sessions ($p<0.001$). This increase remains unaltered even when considering the various vocalization typologies: FB ($p<0.0001$), CT ($p<0.0001$), and PBS ($p<0.0001$). The peak of pulsed activity occurred during PLAY activity.

Fig 6A and Fig 6B depict the density estimation, achieved by applying the kernel density estimation (KDE) method (Gaussian kernel with a bandwidth of 0.4). This method provides the probability distribution of the number of vocalizations (whistles and pulsed vocalizations, respectively) observed within 5-minute time blocks, under two behavioral conditions: organized activity (cyan) and NIGHT (orange) periods.

Practically, the kernel density estimates indicate how frequently specific vocalization counts occur, thereby highlighting differences in the patterns and variability of vocalization output between active and inactive phases. Both density plots reveal distinct patterns in the distribution of vocalizations across the two conditions. The density plot of whistles displays a broader distribution during the organized activity phase, in contrast to the NIGHT period, where vocalizations are densely concentrated around minimal values. The peak of the density curve occurs around 12 whistles per block for the organized activity, while it tends toward much lower values during the NIGHT period. Pulsed vocalizations also exhibit a distribution concentrated around lower values during the NIGHT period (peak at around 5 vocalizations per block, orange curve) compared to the organized activity phase (peak at around 50 vocalizations per block, cyan curve). Moreover, the orange curve declines more steeply than the cyan one, approaching zero shortly after 100 vocalizations per block, whereas the cyan curve decreases more gradually, extending up to 200 vocalizations per block.

The distribution of whistle and pulsed vocalizations across different activity types (ORD, FFR, PLAY, and NIGHT) is visually represented through whisker plots in Fig 7 and Fig 8, respectively. The whistle box plots show distributions for total whistle counts as well as whistle classes based on duration (Class 1–4). The post hoc Dunn test revealed that the median number of total whistles is significantly lower ($p<0.001$) during the NIGHT period than during each one of the other three periods (Fig 7A).

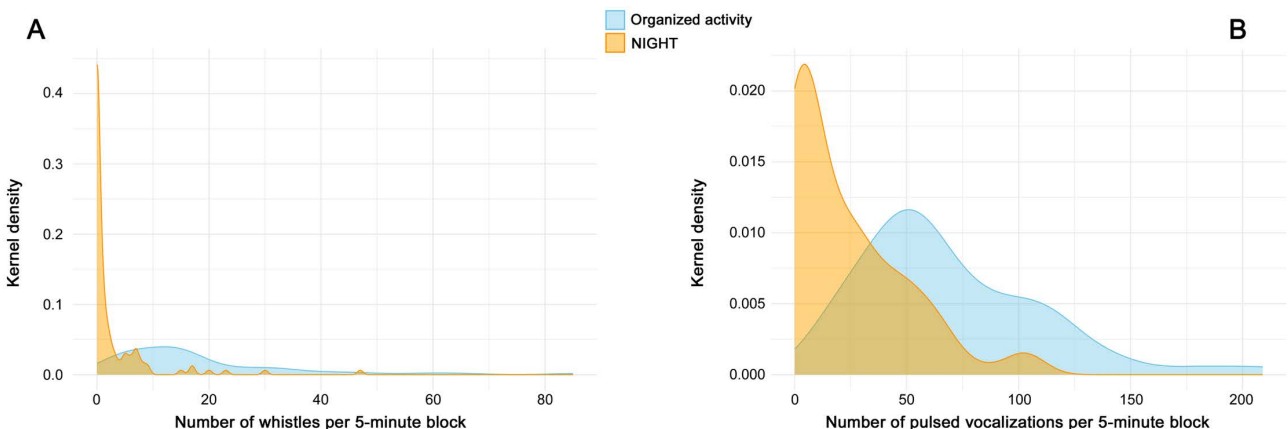

**Fig 6. Kernel density estimation for whistles (panel A) and pulsed vocalization (panel B) distributions.**

No further significant differences (p>0.05) were detected in Fig 7A. This trend is substantially maintained when considering the distinction by whistle duration classes, in particular in the intermediate-duration whistle classes (class 2 in Fig 7C and class 3 in Fig 7D), where the differences between NIGHT and the other three classes are also supported by statistical significance (p<0.001). Regarding pulsed vocalizations, Fig 8 illustrates a gradual increase in the median number of

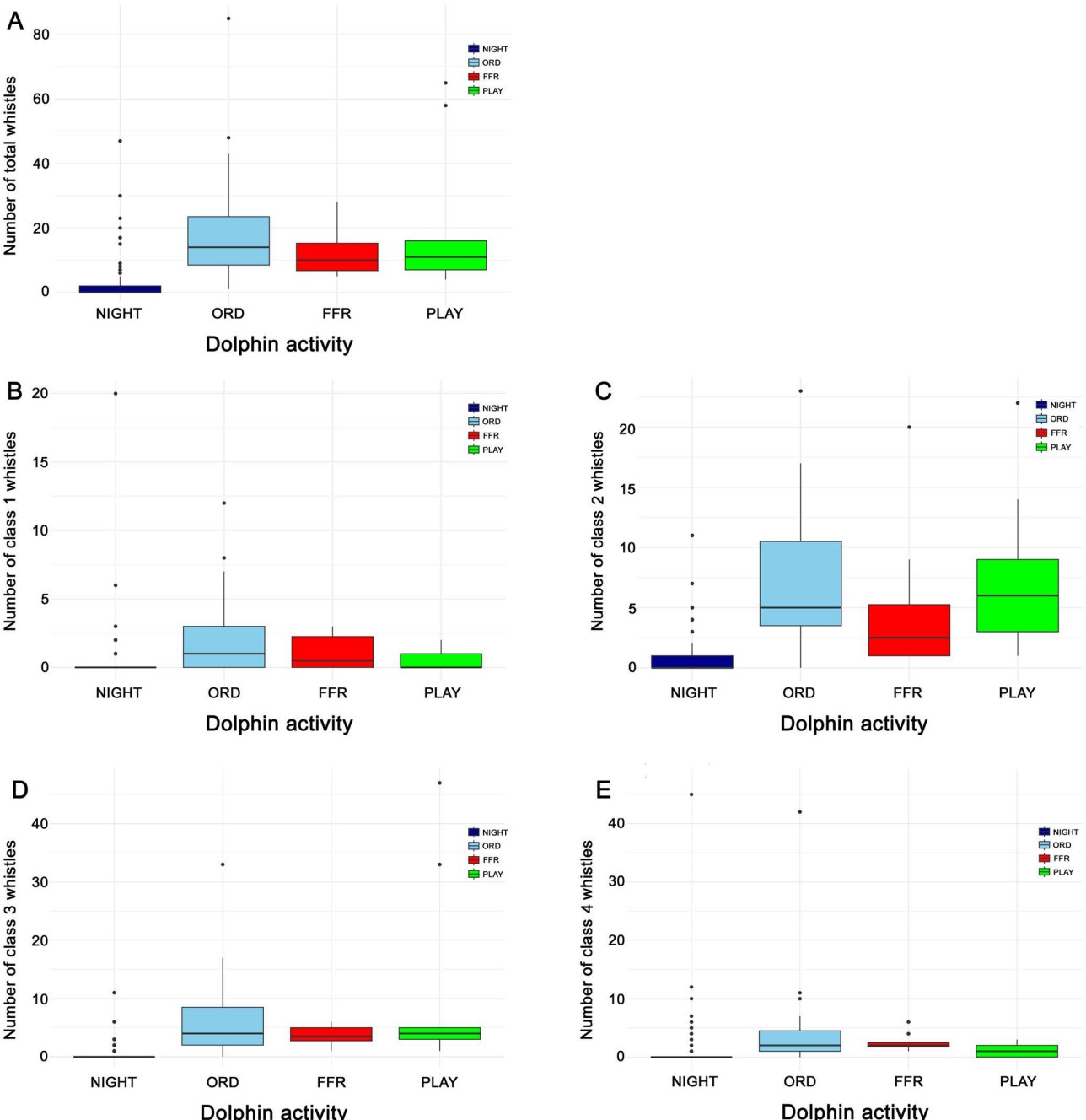

**Fig 7. Whisker plots highlighting the distributions and outliers for each class of whistles in function of the different dolphin activities.** Distribution of total whistles is reported in panel A, class 1 in panel B, class 2 in panel C, class 3 in panel D, and class 4 in panel **E.**

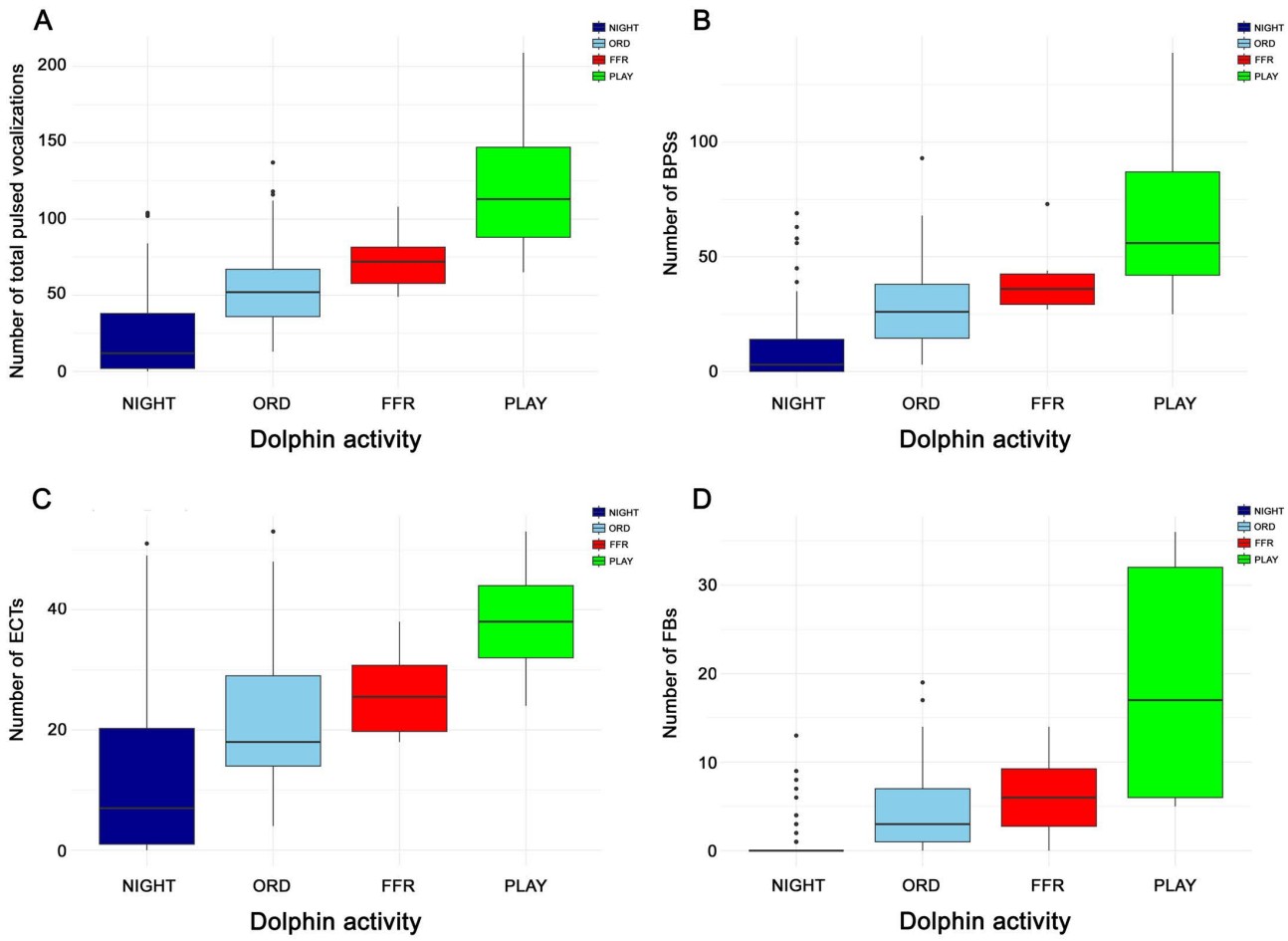

**Fig 8. Whisker plots highlighting the distributions and outliers for each class of pulsed vocalizations in function of the different dolphin activities.** Distribution of total pulsed vocalizations is reported in panel A, BPSs in panel B, ECTs in panel C, and FBS in panel **D.**

occurrences from the NIGHT period to the ORD period, and then to the FFR period, reaching the highest value during the PLAY period in all four graphs (A-D).

Specifically for the total count (Fig 8A), the median number of pulsed vocalizations is significantly lower ($p < 0.001$) during the NIGHT period than during the other periods. This outcome remains statistically significant ($p < 0.001$) even when considering each specific pulsed vocalization, namely ECT (Fig 8B), BPS (Fig 8C), and FB ($p > 0.05$, Fig 8D). Despite the evident increase in the median number of pulsed vocalizations during the PLAY session, no significant differences were found between FFR, ORD, and PLAY sessions for the total count (Fig 8A).

## 4. Discussion

The current study was designed to characterize a newly available dataset of dolphin vocalizations by means of a statistical analysis of the variation of the type and number of vocalizations emitted in function of the dolphin activity during an ordinary day in the marine park. Vocalizations were recorded simultaneously from seven dolphins. Thus, manual extractions of whistles by an experienced PAM analyst, combined with the use of an automated algorithm for identifying pulsed vocalizations, were adopted to minimize potential errors arising from overlapping emissions produced by different

animals. Moreover, it is important to emphasize that the aim of this study was not to quantify the vocal output of each of the seven individual dolphins, but rather to provide an overview of the overall distribution of vocalizations across various activities during a typical day at the marine park.

A key strength of this dataset is its extensive collection of vocalizations, comprising 3,111 whistles, 517 ECT, 489 BPS, and 271 FB (Table 2). The diversity and quantity of signals captured allow us not only to describe vocal output, but also to consider how dolphins may flexibly use different acoustic strategies depending on behavioral contexts. Whistles, often associated with social communication and group coordination, dominate the dataset, while pulsed sounds (ECT, BPS, and FB) appear in contexts linked to exploration, play, and feeding. The presence of feeding buzzes, an often-overlooked vocalization type, further underscores the dataset value in capturing the full range of dolphin acoustic expressions. The large volume of data offers a comprehensive reference for future studies on dolphin vocal modulation and behavioral contexts.

To test the reliability and usability of the current dataset, signal quality was analyzed by evaluating SNR. Table 2 reports the average SNR value for each vocalization type. A widely accepted guideline suggests that an SNR above 10 dB can generally be deemed appropriate for most applications in underwater environments [35]. This threshold means that the signal amplitude surpasses background noise by a factor of at least 10, ensuring a relatively clear distinction between the actual vocalization and unwanted noise components. Among the signals analyzed, only the ECT signals exhibit an average value clearly below 10 dB. Moreover, Table 2 also shows that BPS signals average just below 10 dB, although, considering the standard deviation, a significant portion of these signals exhibit SNR exceeding this threshold. Even though in certain experimental settings SNR values ranging from 5 to 10 dB might still be considered sufficient, caution is advised when interpreting the results obtained with these signals, particularly ECT. Nevertheless, the average SNR across the entire dataset exceeds 11 dB (Table 2).

Statistical analyses were carried out considering the comparisons between the periods of organized activity and the period during which no trainers were present and, consequently, no organized interactions took place. The overall outcomes suggest that all vocalization patterns are markedly influenced by the presence of organized activities, highlighting distinct acoustic behaviors in response to environmental and social contexts. From a functional perspective, significantly higher vocal rates during ORD, PLAY, and FFR sessions can be interpreted as reflecting increased arousal, social coordination, and negotiation among conspecifics, as well as engagement with human trainers. Conversely, the markedly lower rates during NIGHT suggest a context of rest and reduced social demands. Training sessions are based on a stimulus-response framework and therefore always involve human-dolphin interaction. Depending on the specific objective (e.g., group jumping), these sessions may also entail varying degrees of interaction among the animals themselves. Thus, organized activities do not simply increase sound production mechanically, but appear to stimulate communicative functions related to cohesion, synchrony, and response to social and environmental cues.

As mentioned earlier, whistles are the most studied vocalizations of dolphins because they are considered rich in information [6–10,19–23]. In particular, several recent innovative approaches based on artificial intelligence (AI) have been developed for monitoring dolphin presence and behavior [36–40]. These studies have often preferred using whistle spectrograms as input to neural networks, as they are considered more reliable and easier to interpret than other types of vocalizations. Thus, the fact that spectrogram images of 3,111 whistles are made available makes this dataset particularly valuable for bioacoustics research, providing a robust foundation for in-depth analyses of whistle characteristics and variations. Given the large sample size (over 3000 spectrograms), the whistle dataset is of particular interest for AI-based studies. A statistical analysis of the vocalizations was also carried out to highlight the characteristics of the dataset. Specifically, whistles were divided into four classes based on their duration, and the occurrences of each whistle class were statistically characterized across all the activities listed in Table 1. The statistical analysis showed that total whistle occurrences were significantly higher during activity sessions than during NIGHT period ($p < 0.001$). A similar trend was observed across different whistle duration categories. Short-duration whistles (classes 1 and 2), which may function in

rapid exchanges or signaling during close-range interactions, increased during structured sessions, consistent with quick coordination and high arousal states. Longer-duration whistles (classes 3 and 4), potentially linked to maintaining cohesion or signaling persistent intent, also peaked in contexts of high social engagement. This suggests that dolphins flexibly employ different whistle types to serve distinct communicative purposes depending on the context. Comparable trends have also been reported in wild bottlenose dolphins. Quick and Janik (2008) showed that whistle production decreases during coordinated group travel and increases in social contexts, with group size further modulating individual output [41]. This parallels the present findings, where higher vocal activity was observed during structured and socially engaging sessions, and markedly lower rates occurred during night-time, suggesting that the patterns reported here reflect broader species-level tendencies alongside the specific features of the managed-care setting.

The current analysis categorizes whistles solely based on duration. A contour- or identity-based classification was not attempted. Nonetheless, given that the dataset includes recordings from seven individual dolphins, it would be possible to examine the distribution of signature whistles for each dolphin across the day. This, in turn, could allow analogous analyses for non-signature whistles, which may carry context-specific or referential information, as highlighted recently [42]. Such analyses are highly interesting but would require extensive tagging and detailed analysis that go beyond the aims of the present manuscript and will explore in future studies.

A similar analysis was also conducted on the pulsed vocalizations. The total number of pulsed sounds (echolocation clicks, burst-pulse sounds, and feeding buzzes) significantly increased during structured activities compared to free activity periods (NIGHT). Moreover, a statistically significant increase was observed in all subcategories (feeding buzzes, click trains, and burst-pulse sounds) during structured activities compared to NIGHT ($p < 0.001$). Functionally, this pattern seems to confirm that dolphins use pulsed signals to explore, negotiate interactions, and coordinate play or feeding activities. The marked increase of pulsed vocalizations during PLAY sessions is particularly intriguing and may point to communicative functions related to arousal, contact negotiation, or the reinforcement of affiliative bonds. However, the limitations of the present dataset do not allow us to link these signals to specific forms of play or social configurations. Future studies with more detailed ethograms will be needed to clarify these dynamics. Feeding buzzes, though less frequent overall, may play a critical role in coordinating access to food or signaling motivational states.

Further information is provided by the density plots in Fig 6, which reports a detailed visualization of the distribution of vocalization occurrences under different activity conditions. Whistles show a broad distribution, with a higher density of occurrences during structured activity periods (cyan area in Fig 6A), compared to the NIGHT condition, which shows a much narrower distribution with a peak around very low values (orange area in Fig 6A), indicating minimal vocal output. These distributions reinforce the idea that dolphins modulate vocal output as a function of engagement and social needs: structured contexts elicit a broad range of communicative strategies, whereas quiet periods reduce the necessity for vocal coordination. Panel B depicts a similar trend for pulsed vocalizations. The density peak during the NIGHT phase is located at approximately 5 vocalizations per block and declines steeply beyond that. In contrast, the structured activity period shows a peak around 50 vocalizations per block, with a slower decline, extending up to around 200 vocalizations. This suggests that structured sessions not only trigger more frequent signaling but also expand the diversity of communicative functions expressed through vocal behavior. These outcomes align with previous findings that associate structured interactions with increased vocal activity, particularly in training and play sessions [43].

Thus, the present study suggests that dolphins employ vocalizations to meet different communicative needs depending on context, ranging from coordination with conspecifics and trainers, to expressing arousal, to negotiating access to resources. This approach situates the present findings within well-established perspectives in animal communication [44]. While the absence of individual-level tagging and contour-based classification represents a limitation for certain types of analyses, the dataset nonetheless provides important opportunities for future research. The large number of whistle spectrograms, together with the inclusion of pulsed vocalizations and contextual activity information, offers a solid foundation for developing and testing computational approaches, such as artificial intelligence, signal processing, or data

augmentation strategies. In particular, the dataset can be exploited as a training resource for neural networks aimed at detecting dolphin presence, with direct applications in long-term passive acoustic monitoring. By acknowledging both its strengths and its constraints, potential users can better situate their methodological choices and explore diverse applications, from automated signal detection to behavioral context modeling.

Understanding the relationship between vocalization patterns and activity types has direct implications for dolphin welfare and conservation. Present findings seem to suggest that structured activities likely encourage natural behaviors and promote engagement, while ensuring sufficient free activity periods is essential for rest and recovery. This insight can inform the design of enrichment programs to ensure that dolphins in human care maintain a healthy and dynamic acoustic repertoire. Additionally, these findings contribute to broader efforts in marine bioacoustics research by providing a detailed dataset of dolphin vocalizations across multiple contexts. Future research should explore how factors such as group composition, trainer interaction styles, and environmental conditions influence vocal patterns. Applying advanced machine learning techniques to analyze these vocalizations could further refine our understanding of dolphin communication and social behavior.

## 5. Conclusion

The dataset analyzed in this study provides a rich source of information on dolphin vocal behavior in different activity contexts. By integrating expert annotation, spectrogram visualization, and statistical analysis, the dataset allows for a detailed examination of both whistle and pulsed vocalizations. Moreover, the statistical analyses conducted in this study highlight the depth of information made available by this dataset. By quantifying vocalization occurrences across different activity types, the study showcases the utility of detailed acoustic datasets in uncovering meaningful patterns in dolphin communication and behavior. This underscores the importance of making such datasets accessible to researchers and conservationists, as they provide invaluable resources for studying cetacean behavior, developing enrichment strategies, and refining management practices in both captive and wild settings.

## Acknowledgments

The study was made possible through the support and collaboration of Costa Edutainment, which provided access to their Riccione facility. Special acknowledgment is given to Barbara Marchiori, Gianni Bucci, Barbara Acciai, Paola Righetti, and Claudia Di Mecola for their dedicated contributions and support during the project.

## Author contributions

**Conceptualization:** Francesco Di Nardo, Rocco De Marco, David Scaradozzi.

**Data curation:** Laura Screpanti, Francesco Di Nardo, Rocco De Marco.

**Formal analysis:** Laura Screpanti, Francesco Di Nardo.

**Investigation:** Laura Screpanti, Francesco Di Nardo, Rocco De Marco, Stefano Furlati, Giacomo Bucci, David Scaradozzi.

**Methodology:** Laura Screpanti, Francesco Di Nardo, Rocco De Marco.

**Project administration:** Alessandro Lucchetti, David Scaradozzi.

**Resources:** Stefano Furlati, Alessandro Lucchetti, David Scaradozzi.

**Software:** Rocco De Marco.

**Supervision:** Stefano Furlati, Alessandro Lucchetti, David Scaradozzi.

**Visualization:** Giacomo Bucci.

**Writing – original draft:** Laura Screpanti, Francesco Di Nardo.

**Writing – review & editing:** Francesco Di Nardo, Stefano Furlati, Giacomo Bucci, Alessandro Lucchetti, David Scaradozzi.

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
