## [Decision Letter · Decision Letter 0]

31 Jul 2025

PLOS ONE

Dear Dr. Di Nardo,

Thank you for submitting your manuscript to PLOS ONE. After careful consideration, we feel that it has merit but does not fully meet PLOS ONE’s publication criteria as it currently stands. Therefore, we invite you to submit a revised version of the manuscript that addresses the points raised during the review process.

We look forward to receiving your revised manuscript.

Kind regards,

Abebayehu Aticho (PhD, Associate Professor)

Academic Editor

PLOS ONE

 [This work was supported in part by LIFE Financial Instrument of the European Community, Life Delfi Project – Dolphin Experience: Lowering Fishing Interactions (LIFE18NAT/IT/000942) and by the National Recovery and Resilience Plan (NRRP), Mission 4 Component 2 Investment 1.4 (Call for tender No. 3138 of 16 December 2021, rectified by Decree n.3175 of 18 December 2021 of Italian Ministry of University and Research funded by the European Union) NextGenerationEU. The study was made possible through the support and collaboration of Costa Edutainment, which provided access to their Riccione facility. Special acknowledgment is given to Barbara Marchiori, Gianni Bucci, Barbara Acciai, Paola Righetti, and Claudia Di Mecola for their dedicated contributions and support during the project.].

4. Please include a caption for figures 7 and 8.

Additional Editor Comments (if provided):

Reviewers' comments:

Reviewer's Responses to Questions

**Comments to the Author**

1. Is the manuscript technically sound, and do the data support the conclusions?

Reviewer #1: Yes

Reviewer #2: Yes

2. Has the statistical analysis been performed appropriately and rigorously?

Reviewer #1: Yes

Reviewer #2: Yes

3. Have the authors made all data underlying the findings in their manuscript fully available?

Reviewer #1: Yes

Reviewer #2: Yes

4. Is the manuscript presented in an intelligible fashion and written in standard English?

Reviewer #1: Yes

Reviewer #2: Yes

Reviewer #1: Dear Authors

It was an interesting article which presented four classifications of dolphin’s vocalizations in structured and unstructured situations. However, here are some points I hope will contribute to the improvement of your work.

The conclusion presented in the abstract indicates that dolphin vocalizations occur more frequently during structured activities compared to unstructured activities. Can we infer that dolphins interact more with humans (during structured activities) than they do with their peers? Is this due to the presence of humans and their training? Alternatively, does this imply that dolphins interact more with one another during structured activities? I would appreciate your clarification on this matter.

I believe it would be beneficial to provide a brief definition for each technical term mentioned in the text, such as “social and exploratory behaviors”, “social bonding”, “coordination with groups”. Could you clarify what these behaviors refer to?

Were these seven dolphins in the water at the same time, while their sounds were being recorded? or were their vocalizations recorded separately? My understanding from your article is that their sounds were recorded simultaneously. My question is whether the overlap of their vocalizations affects your analysis of their interactive behaviors.

It would be beneficial to provide a brief overview of different types of dolphins in the introduction, and explaining why you chose this particular species (if there is a specific reason for your choice).

I have a similar comment regarding the types of vocalizations (whistles, Echolocation clicks, burst-pulse sounds, feeding buzzes). It would be appropriate to provide a brief definition for each of the four vocalization types in dolphins. Additionally, please specify the frequency range for each type. Do all of them include high-frequency sounds, or are there also vocalizations with lower frequencies among them (particularly those produced using the melon)?

You referred to "trainers" (and not just one trainer). I would appreciate it if you could briefly specify how long these trainers have been interacting with the dolphins, and whether each of them shares an equal level of closeness with the dolphins. Similar to what observed in human infants and children, where the caregiver's relationship can influence the type and frequency of vocalizations, this may also apply to dolphins. For example, the amount of vocalization by a human infant may be greater in the presence of their parents compared to an unfamiliar person/stranger.

What is the purpose of the exercises conducted by the trainers and the tasks assigned to the dolphins? Please provide an example. This question has arisen in my mind: are these exercises intended to enhance the dolphins' vocalizations? For instance, could conditioning and rewarding of dolphins for vocalizing be an integral part of this process?

I am uncertain about the extent to which this suggestion may enhance your work; however, if feasible, it would be more beneficial to compare three situations instead of two:

1. One situation involves various activities (training, FFR, and plays).

2. Another situation pertains to the rest periods between these activities (for example, the interval between the end of a play and the beginning of the next activity).

3. The third situation is nighttime sleep and free activity, which you have considered as a baseline. This is because the amount of vocalizations during the intervals between activities may yield different and insightful data.

Best Regards

Reviewer #2: this paper is a well executed and clearly documented study of dolphin vocalization, offering a valuable dataset and behavioral correlations over a 24-hour period. The technical aspects are sound, as is the labelling, and the results are clearly presented. The authors are committed to data transparency. I encourage publication, but also make some comments which, if addressed, I believe would improve the paper.

- the association of whistle and pulsed vocalization rates with activity contexts is one of the manuscripts main contributions. I would encourage the authors to consider framing these findings more clearly through the lends of behavioral function. For example, increased vocal rates during play or feeding contexts could reflect coordination, arousal, or social negotiation, as some work in dolphin communication explores (e.g. Janik & Slayih, 2013; King et al. 2023). A behavioral framing allows interpretations grounded in what vocalizations do (facilitate cohesion, warn, signal interest), avoid overreaching analogies to linguistic categories. This approach aligns well with well-established views on animal communication (e.g. Seyfarth & Cheney 2010 on primate calls), and would strengthen the paper's discussion on meaning and usage.

- whistles are categorized here solely based on duration (pages 9-10). While this is valid and clear, there is no contour or identity based classification (e.g. signature whistles, shared whistle types), despite the fact that some whistles in the dataset may correspond to such categories. A brief acknowledgement in the discussion of why this was not attempted, or what potential it leaves for follow-up, would help contextualize this choice. as recent work has shown, some non-signature whistles are shared across individuals and can carry context-specific or referential value (e.g. Favaro et al. 2025, King et al. 2023), so this dataset may be of interest for exploring these questions.

- the spike in pulsed vocalizations during the play context is intriguing and could merit deeper reflection. Are these signals potentially linked to rough-and-tumble social play, or associated with particular toy-types or social configurations? While I understand the limitations of the available ethogram, a speculative but information behavioral interpretation (arousal, contact negotiation, affiliate signaling) would help situate these results within the broader literature on dolphin social behavior.

- the authors note that ECTs have average SNR "clearly below" 10 dB, yet Table 2 also shows that BPS averages just below 10 dB. Please clarify further.

- A brief mention of vocal production rates in wild bottlenose dolphins could contextualize the observed patterns and help distinguish between captive-specific behaviors and broader species trends.

minor:

- typo: “Intellegence-based” (p.2, line 40) should read “Intelligence-based.”

- some fine details may be hard to discern at current scale in some figures (especially 4 and 5)

- Adding a short note to guide researchers (especially in AI or signal processing) on how best to engage with the dataset would increase its usability and impact (although this is of course not on the authors, and might be misconstrued as guiding potential users away from limitations if not done neutrally enough)

**Do you want your identity to be public for this peer review?** For information about this choice, including consent withdrawal, please see our Privacy Policy

Reviewer #1: **Yes: ** Mina Fotuhi

Reviewer #2: No

---

## [Author Response · Author response to Decision Letter 1]

14 Oct 2025

Authors thank the Editor and the Reviewers for their constructive criticisms and suggestions. Authors gave careful consideration to the comments and complied with all the points of their critiques, to improve the overall quality of the manuscript. Revisions in the manuscript have been highlighted with yellow color text. To support the revision of the manuscript, six new references have been added in the revised manuscript. The additional journal requirements have been addressed. Specifically, in this revised version of the manuscript, the Authors have updated the data repository following a specific request from the editorial manager. According to the journal’s guidelines, the dataset has now been deposited in SEANOE, one of the repositories listed among PLOS ONE’s recommended options. The data are available under a license fully meeting the journal’s Data Availability requirements.

Reviewer #1:

Comment

Dear Authors

It was an interesting article which presented four classifications of dolphin’s vocalizations in structured and unstructured situations. However, here are some points I hope will contribute to the improvement of your work.

Answer

We sincerely thank the Reviewer for the valuable feedback and for highlighting areas that can help us improve our work. We gave careful consideration to Reviewer’s constructive suggestions and addressed each point to further improve the manuscript. Revisions in the manuscript have been highlighted with yellow color text.

Comment

The conclusion presented in the abstract indicates that dolphin vocalizations occur more frequently during structured activities compared to unstructured activities. Can we infer that dolphins interact more with humans (during structured activities) than they do with their peers? Is this due to the presence of humans and their training? Alternatively, does this imply that dolphins interact more with one another during structured activities? I would appreciate your clarification on this matter.

Answer

This phenomenon of increased vocalizations occurrence during structured activities can be attributed to increased social interactions both with trainers and conspecifics (members of the same species). Training sessions are based on a stimulus-response framework and therefore always involve human-dolphin interaction. This is the reason why dolphins interact more with humans. Depending on the specific objective (e.g., group jumping), these sessions may also entail varying degrees of interaction among the animals themselves. Similarly, in the context of the environmental enrichment program, the nature of the objects introduced to the dolphins can significantly influence their level of interaction. Some items are specifically designed to encourage shared activities, which often involve communication among dolphins, while others are intended for solitary play, aimed at stimulating the abilities of each dolphin individually and are not always accompanied by vocalizations. The Discussion section was redrafted to include this information in the revised manuscript.

Comment

I believe it would be beneficial to provide a brief definition for each technical term mentioned in the text, such as “social and exploratory behaviors”, “social bonding”, “coordination with groups”. Could you clarify what these behaviors refer to?

Answer

We thank the Reviewer for this suggestion. We agree that a brief clarification of these technical terms would improve the clarity of the introduction. In this study, we use these expressions as follows:

• Social and exploratory behaviors refer to activities in which dolphins engage either with conspecifics (members of the same species) or with their surroundings, such as play, object manipulation, or environmental investigation.

• Social bonding describes affiliative behaviors that strengthen relationships among dolphins, including synchronous swimming, tactile contact, and vocal exchanges (e.g., signature whistles) that promote cohesion and trust within the group.

• Coordination with groups refers to the ability of dolphins to align their actions with conspecifics, for instance during cooperative hunting, synchronous movements, or group travel. Vocalizations play a central role in enabling and maintaining such coordinated activities.

As suggested by the Reviewer, in the revised manuscript we have added a brief definition for each of these technical terms into the abstract and the introduction to ensure that readers from different disciplinary backgrounds can follow the description of dolphin vocal behavior.

Comment

Were these seven dolphins in the water at the same time, while their sounds were being recorded? or were their vocalizations recorded separately? My understanding from your article is that their sounds were recorded simultaneously. My question is whether the overlap of their vocalizations affects your analysis of their interactive behaviors.

Answer

Thanks for this observation, which allows us to provide further insight into methodological aspects of the present study. As correctly noted by the Reviewer, dolphins' vocalizations were recorded simultaneously. Manual extractions of whistles by an experienced PAM analyst, combined with the use of an automated algorithm for identifying pulsed vocalizations were adopted to minimize potential errors arising from overlapping emissions produced by different animals. Moreover, it is important to emphasize that the aim of this study was not to quantify the vocal output of each of the seven individual dolphins, but rather to provide an overview of the overall distribution of vocalizations across various activities during a typical day at the marine park. The Discussion section was redrafted to include this further information in the revised manuscript.

Comment

It would be beneficial to provide a brief overview of different types of dolphins in the introduction, and explaining why you chose this particular species (if there is a specific reason for your choice).

Answer

In the Mediterranean Sea, several dolphin species can be found, each with distinct features and behaviors. The Striped Dolphin (Stenella coeruleoalba) is sleek and agile, with gray and white stripes radiating from the eyes, often seen in large pods performing energetic leaps and acrobatics. The Common Dolphin (Delphinus delphis), once widespread but now rare, is recognizable by its striking yellowish hourglass pattern along the sides and its dynamic group behavior. The Bottlenose Dolphin (Tursiops truncatus) is robust and gray with a curved dorsal fin and a short, broad snout, known for its intelligence and frequent interactions with humans. The Risso’s Dolphin (Grampus griseus) stands out with its rounded head and heavily scarred body, which becomes paler with age, and typically lives in smaller, more reserved groups. The bottlenose dolphin is the most commonly represented cetacean species in European zoological facilities. Therefore, the opportunity to record vocalizations was simply taken advantage of in a nearby facility that offered suitable experimental conditions. As suggested by the Reviewer, the Introduction section was redrafted to include this further information in the revised manuscript.

Comment

I have a similar comment regarding the types of vocalizations (whistles, echolocation clicks, burst-pulse sounds, feeding buzzes). It would be appropriate to provide a brief definition for each of the four vocalization types in dolphins. Additionally, please specify the frequency range for each type. Do all of them include high-frequency sounds, or are there also vocalizations with lower frequencies among them (particularly those produced using the melon)?

Answer

Whistles are generally regarded as communicative signals, with their production increasing during social interactions among dolphins. These vocalizations are commonly associated with functions such as individual recognition, coordination of collective behaviors, and regulation of group movements. Typically, they span frequencies from approximately 1 to 25 kHz and exhibit durations ranging from 0.1 seconds to few seconds. In contrast, echolocation clicks are short, broadband acoustic pulses that can reach frequencies up to 140 kHz. These signals play a fundamental role in navigation and prey detection, allowing dolphins to generate detailed auditory representations of their environment. Burst-pulse vocalizations consist of fast sequences of acoustic pulses or clicks with similar characteristics to echolocation clicks. They are distinguished by their high repetition rate and variable spectral characteristics. These vocalizations are believed to serve social functions and are frequently observed in contexts involving agonistic behavior, such as during competitive feeding events. Feeding buzzes are rapid sequences of pulses emitted by dolphins as they close in on prey. These high-rate pulse trains occur during the final phase of prey pursuit, when the animal requires fine-scale spatial information to catch its moving target. For bottlenose dolphins, dominant click frequencies typically fall between 40 and 80 kHz. As suggested by the Reviewer, the Introduction section was redrafted to include this further information in the revised manuscript.

Comment

You referred to "trainers" (and not just one trainer). I would appreciate it if you could briefly specify how long these trainers have been interacting with the dolphins, and whether each of them shares an equal level of closeness with the dolphins. Similar to what observed in human infants and children, where the caregiver's relationship can influence the type and frequency of vocalizations, this may also apply to dolphins. For example, the amount of vocalization by a human infant may be greater in the presence of their parents compared to an unfamiliar person/stranger.

Answer

At the time of the study, all trainers at Oltremare possessed substantial professional experience, ranging from a minimum of 5 up to 10 years. The dolphin group housed at the facility had remained stable for several years, ensuring consistency in social dynamics and behavioral baselines. The operational protocol at Oltremare mandates that all trainers engage uniformly with each individual dolphin, a practice specifically designed to prevent the emergence of preferential bonds and to maintain balanced human-animal interactions across the group. The Data collection section was redrafted to include this information in the revised manuscript.

Comment

What is the purpose of the exercises conducted by the trainers and the tasks assigned to the dolphins? Please provide an example. This question has arisen in my mind: are these exercises intended to enhance the dolphins' vocalizations? For instance, could conditioning and rewarding of dolphins for vocalizing be an integral part of this process?

Answer

The main objective of the exercises conducted by trainers and the tasks assigned to the dolphins is to promote both physical and cognitive stimulation, as well as to enhance the variability of the controlled environment, which is inherently less stimulating than the natural habitat. These activities are not specifically aimed at enhancing vocal behavior. In particular, vocal conditioning and the reinforcement of vocalizations are not considered integral components of the training protocol. The Data collection section was redrafted to include this information in the revised manuscript.

Comment

I am uncertain about the extent to which this suggestion may enhance your work; however, if feasible, it would be more beneficial to compare three situations instead of two:

1. One situation involves various activities (training, FFR, and plays).

2. Another situation pertains to the rest periods between these activities (for example, the interval between the end of a play and the beginning of the next activity).

3. The third situation is nighttime sleep and free activity, which you have considered as a baseline. This is because the amount of vocalizations during the intervals between activities may yield different and insightful data.

Answer

We appreciate the Reviewer’s thoughtful suggestions, which may open up a new direction for future investigations. However, such analyses would require further extensive tagging and detailed analysis which go beyond the aims of the present manuscript. We will definitely take this suggestion into account in our future research.

Reviewer #2:

Comment

this paper is a well executed and clearly documented study of dolphin vocalization, offering a valuable dataset and behavioral correlations over a 24-hour period. The technical aspects are sound, as is the labelling, and the results are clearly presented. The authors are committed to data transparency. I encourage publication, but also make some comments which, if addressed, I believe would improve the paper.

Answer

We thank the Reviewer very much for the thoughtful and encouraging comments, which are helpful not only for improving the present study but also for outlining a path for potential future work. We gave careful consideration to Reviewer’s constructive suggestions and addressed each point to further improve the manuscript. Revisions in the manuscript have been highlighted with yellow color text.

Comment

- the association of whistle and pulsed vocalization rates with activity contexts is one of the manuscripts main contributions. I would encourage the authors to consider framing these findings more clearly through the lends of behavioral function. For example, increased vocal rates during play or feeding contexts could reflect coordination, arousal, or social negotiation, as some work in dolphin communication explores (e.g. Janik & Slayih, 2013; King et al. 2023). A behavioral framing allows interpretations grounded in what vocalizations do (facilitate cohesion, warn, signal interest), avoid overreaching analogies to linguistic categories. This approach aligns well with well-established views on animal communication (e.g. Seyfarth & Cheney 2010 on primate calls), and would strengthen the paper's discussion on meaning and usage.

Answer

We thank the Reviewer for this valuable suggestion. In the revised version of the manuscript, we have taken this advice into account and re-framed the Discussion and by interpreting the results more explicitly through the lens of behavioral function. The reference suggested by the Reviewer was also included [44].

Comment

- whistles are categorized here solely based on duration (pages 9-10). While this is valid and clear, there is no contour or identity based classification (e.g. signature whistles, shared whistle types), despite the fact that some whistles in the dataset may correspond to such categories. A brief acknowledgement in the discussion of why this was not attempted, or what potential it leaves for follow-up, would help contextualize this choice. as recent work has shown, some non-signature whistles are shared across individuals and can carry context-specific or referential value (e.g. Favaro et al. 2025, King et al. 2023), so this dataset may be of interest for exploring these questions.

Answer

We thank the Reviewer for this insightful suggestion, which provides a clear and novel direction for a future study focused specifically on this question and potentially a follow-up publication. We agree with the Reviewer that, given that our dataset includes recordings from seven individual dolphins, it would be possible in future work to examine the distribution of signature whistles for each dolphin across the day. This, in turn, could allow analogous analyses for non-signature whistles, which may carry context-specific or referential information, as highlighted by recent studies indicate by the Reviewer. Such analyses are highly interesting but would require further extensive tagging and detailed analysis which go beyond the aims of the present manuscript. This point has now been explicitly included in the revised manuscript.

Comment

- the spike in pulsed vocalizations during the play context is intriguing and could merit deeper reflection. Are these signals potentially linked to rough-and-tumble social play, or associated with particular toy-types or social configurations? While I understand the limitations

---

## [Editor Report · Decision Letter 1]

27 Oct 2025

Acoustic Analysis of Bottlenose Dolphin Vocalizations for Behavioral Classification in Controlled Settings

PONE-D-25-34379R1

Dear Dr. Francesco, 

We’re pleased to inform you that your manuscript has been judged scientifically suitable for publication and will be formally accepted for publication once it meets all outstanding technical requirements.

Kind regards,

Abebayehu Aticho

Academic Editor

PLOS ONE
---

## [Editor Report · Acceptance letter]

PONE-D-25-34379R1

PLOS ONE

Dear Dr. Di Nardo,

I'm pleased to inform you that your manuscript has been deemed suitable for publication in PLOS ONE. Congratulations! Your manuscript is now being handed over to our production team.

Kind regards,

on behalf of

Professor Abebayehu Aticho

Academic Editor

PLOS ONE